# SANTé: A Light-weight End-to-End Semantic Search Framework for RDF data

Edgard Marx,[1,2][0000−0002−3111−9405] André Valdestilhas,[1]
Hannah Beck,[2] and Tommaso Soru[1]

[1] AKSW, Institute of Computer Science, University of Leipzig
{marx,valdestilhas,soru}@informatik.uni-leipzig.de
[2] Leipzig University of Applied Science (HTWK)
{edgard.marx,hannah.beck}@stud.htwk-leipzig.de

**Abstract.** Natural language interfaces are one of the most powerful technologies to enable content access. It is a diverse and thriving topic that tackles a multitude of challenges ranging from designing better ranking models to user interfaces. Developing or adapting search engines is a very time-demanding and resource-consuming task. We present `SANTé`, a semantic search framework that facilitates publishing, querying, and browsing RDF data sets. We show the different interfaces implemented by `SANTé` through guided steps from raw RDF data to the search result using keyword queries. We demonstrate how `SANTé` can be used to publish and consume RDF data.

**Repository**: `http://github.com/AKSW/sante`

**License**: `https://www.apache.org/licenses/LICENSE-2.0`

**FOAF demo**: `http://foaf.aksw.org/`

**Pokémon demo**: `http://pokemon.aksw.org/`

## 1   Introduction

There is an enormous amount of machine-readable data published on the Web ranging from a variety of serialization formats and domains. Among the most used serialization formats lies the W3C standard Resource Description Framework (RDF).[3] RDF advocates for a flexible-schema approach that allows publishers to curate content (re-)using self-descriptive metadata. Many institutions such as Google[4] and the German National Library[5] have adopted the W3C standard either for consuming or publishing information. To date, over 600 thousand RDF data sets [8] are openly accessible on the Web over interfaces that facilitate its access such as SPARQL[6] and Comunica [7]. However, most of these initiatives

---

[3] `https://www.w3.org/RDF`

[4] `https://developers.google.com/search/docs/data-types/product`

[5] `https://wiki.dnb.de/pages/viewpage.action?pageId=68060017`

[6] `https://www.w3.org/TR/sparql11-query`

require lay users to be familiar with RDF standards and domain-specific languages. Additionally, many of the RDF data available on the Web has no equivalent human-friendly format such as web pages or relies on third-party search engines such as Google for content access and discovery. Over the last years, several approaches such as question answering [2], search [4] and user interfaces [1] have been proposed to address this problem. In this article, we demonstrate SANTé, an open-source semantic search framework that aims to democratize RDF access by providing an end-to-end semantic search framework. SANTé is a result of several years of research [4,5] and is designed for enabling RDF data publishing, browsing, and search through keyword queries. SANTé can be used to leverage complex applications such as SPARQL query building capabilities using natural language queries [3] and facet search [6]. In this work, we show SANTé's different built-in functionalities and demonstrate how to publish arbitrary RDF data in the following section. We conclude with an outlook on future work.

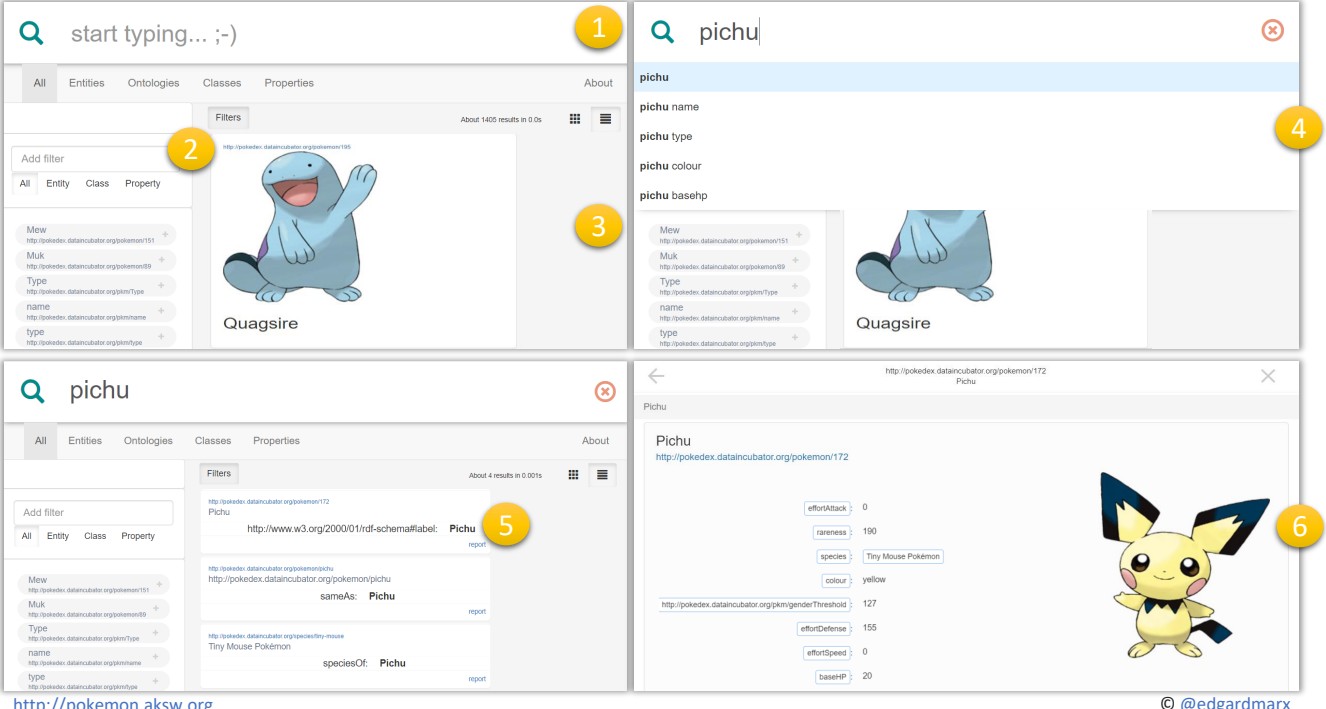

**Fig. 1.** An overview of six different features available in SANTé User Interface over the Pokémon data set: ①) Search bar; ②) Faceted Filter: enables to refine the search and to perform faceted navigtion through the addition of graph pattern based filters; ③) Knowledge cards: simplifies the information visualization; ④) Autocomplete: offers automatic suggestions based on the user's query; ⑤) Structured Highlights: highlights the search result accordingly to the best match property-object(s) and generates concise snippets; ⑥) Data browser: allows to explore and browse content and search results.

## 2   Demonstration

The goal of this demonstration is to cover the necessary steps of making an arbitrary RDF data set accessible using keyword queries. We showcase a practical example of instantiation using standard ontologies. We aim to promote a community discussion around the topic and to gather relevant feedback. In the following, we provide a guided outline of the publishing pipeline and access interfaces. `SANTé`'s code and releases are openly available at `https://github.com/AKSW/sante`. To facilitate the assessment and evaluation, a short animation and video demonstrating `SANTé`'s capabilities are also available on the Git repository.

### 2.1   Indexing & Instantiating

The RDF framework allows users to model concepts and their relations in a structured manner. Ontologies such as OWL and RDFS are powerful tools for creating metadata. One of the most distinguishable RDF features is the possibility of using reasoners to infer unexplicit hierarchies and relations. `SANTé` relies on triple stores for index creation, which can support different levels of reasoning. In the following running example, we show how to instantiate a KBox[7] endpoint containing the FOAF ontology and its dependencies (Listing 1.1) as well as how to create an index from there using a command line (Listing 1.2).

```
1 java -jar kbox.jar -server -kb "http://xmlns.com/foaf/0.1,
    https://www.w3.org/2000/01/rdf-schema,http://www.w3.org
    /2002/07/owl,http://www.w3.org/1999/02/22-rdf-syntax-ns,
    http://purl.org/dc/elements/1.1/,http://purl.org/dc/terms
    /,http://purl.org/dc/dcam/,http://purl.org/dc/dcmitype/"
    -install
2 Loading Model...
3 Publishing service at http://localhost:8080/kbox/sparql
4 Service up and running ;-) ...
```

**Listing 1.1.** Instantiating an endpoint using FOAF ontology and its dependencies.

```
1 java -jar sante-vXXX.jar index -endpoint http://localhost
    :8080/kbox/sparql -path \foaf
```

**Listing 1.2.** Indexing the FOAF ontology and its dependiencies instantiated in Listing 1.1.

After indexing, the content can be published using `SANTé`'s Web Service `WAR` file as follows.

```
1 java -jar sante-vXXX.jar server -war sante-vXXX.war -path \
    foaf -port 9090
```

**Listing 1.3.** Instantiating `SANTé`'s webserver with the FOAF ontology and its dependencies previously indexed (see Listing 1.2).

---

[7] `https://github.com/AKSW/KBox`

If all steps above have been successfully followed, the Web search interface will be accessible at `http://localhost:9090`.

**Searching & Browsing** Figure 1 gives an overview of `SANTé`'s search and browsing capabilities. It is possible to refine the search with graph pattern filters or explore and navigate through the metadata over user-friendly web pages. `SANTé` works with customizable Knowledge Cards. Knowledge Cards are *rich cards*[8] that contain useful information about something and could be enriched with links, pictures, and other types of media accordingly to the necessity (③ in Figure 1). Another `SANTé`'s feature is dubbed *Structured Highlights* (⑤ in Figure 1). Common search engines display results using `feature snippets` and `OneBox results`.[6] They present relevant web page text blocks in case of the former or an inline answer in case of the latter. Structured Highlights are knowledge-card-snippets automatically generated using the most likely property-objects containing the information sought. Structured Highlights works as a cognitive activity snapshot giving an outlook on every available relevant information through highly activated graph connections—using *P [4].

## 2.2   Access Interfaces

To facilitate integration and information consumption, `SANTé` allows to search through four different `REST` APIs and a command-line interface:

**/API/lookup** exposes a `JSON` REST interface that allows to access the indexed data using the DBpedia `lookup` API.[9]

**/API/reconcile** implements the Reconcile Service API Specification Version 0.1.[10] with limited support to `queries`[11] over `HTTP GET`.

**/API/search** *and* **/API/suggest** exposes resp. the search and auto-suggestion REST APIs, allowing to restrict results by class, URI- and URI-prefixes.

**Command-line interface** In addition to the four `REST` interfaces, it is also possible to search using a `command-line` interface as follows:

```
java -jar sante-vXXX.jar -query "resource" -path \foaf
```

**Listing 1.4.** Searching for all occurrences of the word "*resource*" in the FOAF ontology.

---

[8] https://developers.google.com/search/docs/advanced/appearance/
   search-result-features
[9] https://wiki.dbpedia.org/lookup
[10] https://www.w3.org/community/reconciliation/
[11] https://reconciliation-api.github.io/specs/0.1/#reconciliation-queries

### 2.3   Showcases

`SANTé`'s different capabilities are showcased in two live instances:

- `http://foaf.aksw.org/` This is the live instance of the running example presented in this paper. The user can experience a real-time search where the result is computed while the query is being typed. It showcases SANTé's simple (search and data browser) interface on publishing the FOAF ontology.
- `http://pokemon.aksw.org/` This instance showcases SANTé's full functionalities (search, autocomplete, REST APIs, facet search using graph pattern based filters, and data browser) over the Pokémon data set.

## 3   Conclusion

In this work, we presented an open-source framework that enables publishing, browsing, and search RDF data through keyword queries. The presented framework is designed to facilitate lay users to access RDF data contents. The next efforts will consist of: (1) Facilitating content extraction, streaming, and access with query languages; (2) Improving the user interfaces; Integrate (3) entity recommendation, (4) versioning, and, (5) content curation. We see this work as the first step towards human- and machine-enabled content access. We are looking forward to fruitful collaborative engagement with RDF data set publishers and consumers.

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
