# OpenReview forum: "SANTe ́: A Light-weight End-to-End Semantic Search Framework for RDF data"
_eswc-conferences.org/ESWC/2021/Conference/Poster_and_Demo_Track — ESWC2021 P&D_

### Official Review · ~Yoan_Chabot1 · 2021-04-12
**Interesting tool to hide the complexity of RDF data but difficult to see the contributions of SANTé compared to existing tools**

**Rating:** 5
**Confidence:** 3

**Review:**

The paper proposes a demonstration of SANTé, a semantic search framework to facilitate the publication, query and browsing of RDF datasets.
This is an interesting problem with an important issue: the adoption of semantic technologies by non-expert user communities.

The code of the tool is available at https://github.com/AKSW/sante.
The documentation provided is clear and allows either to create an index or to test the tool on pre-existing datasets (FOAF and Pokémon).
The documentation effort of the solution is appreciable.

The tool offers an ergonomic interface that perfectly hides the dryness and complexity of the RDF format.
It is particularly easy to browse the RDF dataset or make queries via the search field.

That being said, I find that the way filters work is not clear.

Taking the "Pokémon" demo as an example, shouldn't a search using the search bar for "electric type" then appear in the filters?
Is it possible to add this "Electric Type" filter via the widget on the left of the application? I found how to add a "type" filter but it seems impossible to set a value.
Neither the documentation nor the paper can clarify these points.
It would be nice to add more details on how the "Entity", "Class" and "Property" filters work as each user may have a different intuition on these concepts.

At the beginning of the paper, the authors insist on the importance of natural language interfaces.
They are indeed of great importance, especially for users who are not familiar with standards such as RDF and SPARQL.

One of the main problems is that the proposed demonstration does not highlight the tool's ability to process natural language.
The proposed examples and the demonstration do not allow to appreciate the capabilities of SANTé on these aspects.
The authors should rework the demonstration in this direction or put less emphasis on these aspects in the abstract/intro.
Moreover, it would be appreciated if the authors could position SANTé in relation to similar tools like SPARKLIS.
This last tool, according to me, offers more advanced functionalities and allows to reduce even more the gap between semantic technologies and the non-expert user by using natural language in a more prominent way.
* Sparklis: An expressive query builder for SPARQL endpoints with guidance in natural language
* SPARKLIS: a SPARQL endpoint explorer for expressive question answering

It would be interesting to know the point of view of the authors on SPARKLIS and the other works of the research community.

In the APIs section, the authors say that "SANTé implements five APIs" but only 4 APIS are presented.

The paper is clear and well organized. The same goes for the readme on https://github.com/AKSW/sante.
However, there are many errors and I can only advise the authors to proofread the whole paper and this readme.
There is a repetition on page 3: "SANTé relies on triple stores for index creation. SANTé relies on trile stores for index creation..."
The sentence in the conclusion "was developed during several research years" repeats with the introduction.

Typos :
* Page 2: "pratical"
* Page 2: "instantion"
* Page 2: "releases are open available"
* Page 3: "distiguishable"
* Page 3: "hyerarchies"
* Page 4: "thorugh"
* Page 4: "pipe-line"

Typos in the readme:
* "instatiate" x 4
* "informaiton"
* "availabes"
* "streigh forward"
* "That's usefull"
* "availabes"
* "of the work pokemon"

**Anonymity:**

No, I would like my review to be deanonymized.

---

### Official Review · ~David_Chaves-Fraga1 · 2021-04-14
**Very interesting tool useful for non-semantic web users**

**Rating:** 7
**Confidence:** 5

**Review:**

This paper presents an open-source tool to browse, and search RDF data. These kinds of tools are very interesting and needed in the semantic web community to remove the barriers of adoption by other communities. As the authors mention, it is supported by many years of research. The tool provides documentation enough and some tutorials to start to work with it. Main comments:
- Please provide a License to the code in the Github repository. Additionally, following good open science practices, you can connect Zenodo and Github to provide a DOI to each release of your code.
- Are the listings really needed in the paper? I would use that space to incorporate more useful information to the readers (for example, give a bit more detail of the (semantic web) technologies used under the frontend) and, delegating the technical part to the readme of the Github
- Would be great to have an instance of the tool publicly accessible to test it.


**Anonymity:**

No, I would like my review to be deanonymized.

---

### Official Review · AnonReviewer4 · 2021-04-15
**review of Santé A Light-weight End-to-End Semantic Search Framework for RDF data**

**Rating:** 4
**Confidence:** 5

**Review:**

In this demo/poster the authors present a system that loads and indexes RDF data and allows to visualize these data. However I see many similarities with other systems such as Graph [1,2]. I know there is not much space in a poster paper, however one paragraph indicating that there are other systems and why the author's approach is new/better/different would have sufficed.

[1] José Moreno-Vega, Aidan Hogan: GraFa: Faceted Search & Browsing for the Wikidata Knowledge Graph. International Semantic Web Conference (P&D/Industry/BlueSky) 2018
[2] 	José Moreno-Vega, Aidan Hogan: GraFa: Scalable Faceted Browsing for RDF Graphs. International Semantic Web Conference (1) 2018: 301-317


**Anonymity:**

Yes, I would like my review to remain anonymous.

---

### Official Review · Program_Chairs · 2021-04-19
**Metareview: Accept (unclear novelty, lack of public demo but promising tool)**

**Rating:** 6
**Confidence:** 5

**Review:**

The paper has received positive comments, but mixed reviews. On the one hand the reviewers acknowledge the importance of such tools towards making the Semantic Web more usable and more accessible to a broader range of users. On the other hand, they also state that the novelty of the work is unclear, that the system lacks a public demo, that it is not clear how natural language plays a role, and that some of the functionalities are not so intuitive. Overall, given that the work is promising and could prove useful, and that the P&D session historically welcomes discussion on ongoing work in which authors can seek feedback, use-cases, etc., we think that the paper could be accepted, but request that the authors make clear the current limitations and clarify their novelty with respect to existing works.

**Anonymity:**

Yes, I would like my review to remain anonymous.

---

### Decision · Program_Chairs · 2021-04-19

Accept